# First Human Case of Tick-Borne Encephalitis in Non-Endemic Region in Italy: A Case Report

**DOI:** 10.3390/pathogens11080854

**Published:** 2022-07-29

**Authors:** Nicole Barp, Cinzia Cappi, Marianna Meschiari, Marzia Battistel, Maria Vittoria Libbra, Maria Alice Ferri, Stefano Ballestri, Altea Gallerani, Filippo Ferrari, Marisa Meacci, Mario Sarti, Mariano Capitelli, Cristina Mussini, Erica Franceschini

**Affiliations:** 1Department of Infectious Diseases, University of Modena and Reggio Emilia, 41124 Modena, Italy; alteagallerani@gmail.com (A.G.); cristina.mussini@unimore.it (C.M.); 2Department of Internal Medicine, Pavullo Hospital, 41124 Modena, Italy; c.cappi@ausl.mo.it (C.C.); m.libbra@ausl.mo.it (M.V.L.); ma.ferri@ausl.mo.it (M.A.F.); s.ballestri@ausl.mo.it (S.B.); m.capitelli@ausl.mo.it (M.C.); 3Department of Infectious Diseases, AOU Modena, 41124 Modena, Italy; mariannameschiari1209@gmail.com (M.M.); ericafranceschini0901@gmail.com (E.F.); 4Department of Laboratory Medicine, San Martino Hospital, ULSS1 Dolomiti, 32100 Belluno, Italy; marzia.battistel@aulss1.veneto.it; 5Department of Microbiology, AOU Modena, 41124 Modena, Italy; ferrari.filippo@policlinico.mo.it (F.F.); meacci.marisa@policlinico.mo.it (M.M.); sarti.mario@aou.mo.it (M.S.)

**Keywords:** tick-borne encephalitis, tick-borne encephalitis virus, ticks-borne disease, viral encephalitis

## Abstract

Tick-borne encephalitis (TBE), a human viral infectious disease caused by the tick-borne encephalitis virus (TBEV), is emerging in Italy, especially in the north-eastern area. No human cases of autochthonous TBE have been reported in Italy’s central regions (such as Emilia-Romagna, Italy). However, here we describe the first human case of TBEV infection in this region, pointing to endemic transmission of TBEV, supporting the concept of circulation of TBEV and of the presence of a possible hot spot in the Serramazzoni region in the Emilian Apennines.

## 1. Introduction

Tick-borne encephalitis (TBE) is a human viral infectious disease caused by tick-borne encephalitis virus (TBEV), a member of the *Flaviviridae* family, and is usually transmitted by tick bites, especially *Ixodes ricinus*. Three main virus subtypes are described: European or western tick-borne encephalitis virus (TBEV-Eu); Siberian tick-borne encephalitis virus; and far eastern tick-borne encephalitis virus. In Europe, including the Czech Republic, Estonia, Latvia, Lituania, Poland, Switzerland, the southern part of Scandinavian Peninsula, the northeastern part of Croatia, the southern part of Germany, the western and northern part of Hungary and some districts of Austria, the TBEV-Eu infection is endemic. In these areas, *Ixodes ricinus* act both as vector and reservoir. The infection in humans may occur with a monophasic (with or without neurologic symptoms) or biphasic course (first stage with no-specific symptoms, asymptomatic interval, and second stage with neurological involvement) [1,2]. During the first stage, leukopenia as well as thrombocytopenia and abnormal liver function are found. During the second phase, white blood cell count and C-reactive protein (CRP) may be elevated. Analysis of cerebrospinal fluid (CSF) usually shows pleocytosis and a moderately raised protein level [2,3,4]. The European Union’s case definition of TBE is based on symptoms of inflammation of the Central Nervous System and one of the following laboratory confirmations: TBEV specific IgG and IgM antibodies in blood, TBEV specific IgM antibodies in CSF, seroconversion or fourfold increase of TBE specific antibodies in paired serum samples, detection of TBEV viral nucleic acid, or isolation of TBEV from clinical specimen [5]. In Italy, TBEV-Eu infection is endemic in three North-Eastern areas: Belluno, Trentino-Alto Adige and Friuli-Venezia Giulia [2]. No human cases of autochthonous TBE have been reported in Emilia Romagna, Italy.

## 2. The Case Description

On 12 April 2022, a 59-year-old obese man (body max index of 30.03) presented in Pavullo Hospital with complaints of fever (temperature 41 °C), headache, and arthromyalgia. He had no comorbidities, no chronic therapies, and he did not remember any tick bites; he was not vaccinated for TBE or Yellow Fever and he had not pets at home. He had not consumed goat-milk and he had not travelled outside Serramazzoni, his home town, in the two months before becoming ill. However, he went trekking in bushy and forested areas in this district. His clinical history was characterized by two phases (Table 1). The first one appeared on 29 March and was characterized by fever and asthenia for four days; after the asymptomatic period (3 April 2022–11 April 2022), he started to present symptoms typical of the second stage as high fever (temperature 41 °C), headache, ideomotor slowing, ataxia, constipation, and dyspnea. Due to the severity of the clinical picture he was admitted to hospital. A total body computed tomography scan showed no abnormalities. Blood exams reported only neutrophil leukocytosis. We performed a lumbar puncture; CSF was clear, presenting high levels of proteins (57 mg/dL, cut off 20–50 mg/dL) and 60 lymphocytes (cut off <4).

After lumbar puncture, empiric antibiotic, and antiviral therapy (ceftriaxone and acyclovir) were administered. Initially low-flow oxygen therapy was necessary, since oxygen saturation was 94% and the patient had dyspnea. Cardiac frequency was normal, while blood pressure values were tendentially high, so an antihypertensive therapy with ACE-inhibitor was started. After a few days, limb tremors and left facial nerve palsy appeared. Then, fever disappeared from the following day. Concerning blood exams, neutrophil leukocytosis persisted, later associated with monocytosis, slightly high levels of CRP (about 1 mg/dL), hyponatremia, hypokalemia, and liver elevation enzymes (especially ALT). The most common differential causes of encephalitis (Cytomegalovirus, Epstein Barr Virus, Herpes Virus 1 and 2, Adenovirus, Enterovirus, Toscana virus, different bacterial infections) were excluded through PCR in CSF. West Nile virus was not tested since in Emilia Romagna the virus is usually tested only during the period of increased vector activity (May-November) and was not already been detected in sentinels (birds) in that period of the year [6].

Since the patient lives in a Lyme disease (LD) endemic region, IgG and IgM for *Borrelia burgdoferi* (Bb) in serum and in CSF were investigated, using chemiluminescent immunoassay (CLIA) and immunoblot. In serum IgG (CLIA 240 AU/mL, cut off 15 AU/mL) were positive for antigens p83, p39, p30, p21, Osp17/DbpA, p14 and VlsE, negative for OspC, p58 and p43, while IgM were positive for antigens Osp17, VlsE and p41, negative for OspC and p39. In CSF IgG (CLIA 26.5 AU/mL, cut off 5.5 AU/mL) were positive for antigens Osp17, p14, VlsE, negative for p83, p58, p43, p39, p30, OspC, p21, DbpA while IgM were totally negative; PCR for Bb in CSF was negative.

On 21 April (nine days from the admission), serology for TBEV both in serum and in CSF was performed: in serum IgG (1651 VIEU/mL, cut off >165 VIEU/mL) and IgM (6 VIEU/mL, cut off >1 VIEU/mL) ELISA were positive; in CSF, positivity of IgG and IgM ELISA were confirmed also with immunochromatographic method. It was not possible to perform PCR for TBEV in CSF due to improper CSF storage [7].

Since the patient was never treated for Bb before, empiric antibiotic therapy was confirmed. Treatment for TBE was only supportive. Patient continues to present asthenia and tremors after a month from the infection.

## 3. Discussion

This is the first autochthonous case of TBE in Serramazzoni, a mountain district of Modena (Emilia-Romagna) in the Apennines, and thus in a central region of Italy. In this area, the climate and availability of hosts are particularly suitable for the development of ticks and the maintenance of the life cycle of tick-borne pathogens. Summers are warm and sunny, while winters are cold and snowy. During the year, the temperature generally ranges from −1 °C to 26 °C, rarely below −6 °C or above 30 °C. Average humidity is 75%, while rainfall ranges from about 78 mm in October to about 25 mm in January. In relation to topography, there are significant altitude variations (from 162 m to 906 m above sea level), with an average altitude of 669 m; Serramazzoni is covered by tree areas (48%), cultivated land (46%) and different torrents [8,9]. While bacteria belonging to genus *Anaplasma, Rickettsia, Babesia* and *Borrelia* were found in ticks (especially in *Ixodes ricinus*), TBEV-positive ticks had not been found in this district [10]. Ticks which transmitted TBEV may have been transported by birds or other animals from TBEV endemic regions (Figure 1), since the distance between Serramazzoni and the closest endemic area (Belluno), is about 300 km. Defining a TBE risk area in the emerging phase of the disease is difficult [11]; TBE is a preventable disease, and efficacious vaccines against TBEV are available. WE maintain that they should be administered to population in risk areas [12].

Our patient had not reported any tick bites, but tick-borne diseases should be considered for diagnosis, since 50% of patients with tick-borne disease ignore tick bites due to the modulation of pain reflexes and itching of the active molecules in tick-saliva [2].

Our patient clinically presented a typical biphasic course of TBE: the aforementioned first episode with fever may represent the first viremic phase of the infection, usually characterized by non-specific symptoms [1,2,13]. The asymptomatic interval (nine days) was followed by a second phase with neurologic symptoms [2,14].

Concerning laboratory features, typical infection and inflammation features were present; CRP levels were slightly raised; and only 50% of patients with TBE presented high CRP levels. The diagnosis of TBEV infection is not to be excluded if CRP is in range. During TBE, liver involvement could be indicated by elevation of liver enzymes. Electrolyte disorders could be caused by fever with sweating [1,2,12]. Finally, our patient had typical CSF features of TBE, namely high protein levels and the presence of lymphocytes [1,2].

IgG and IgM in serum and IgG in CSF for Bb were detected. It was difficult to distinguish between a primary infection and a previous infection. The positive detection of Bb IgM and IgG alone is not an indication of an acute illness, since elevated IgG and IgM antibody titles in serum or in CSF are not uncommon for years in healthy individuals following Bb contact [15]. While positivity of IgM against OspC, p41 and VlsE show an early immune response, IgG against p83, p58, p43, p39, p30, p21, Osp17/DbpA, p14 and VlsE show a late immune response. In addition, intrathecal IgM synthesis occurs in 80–100% during neuroborreliosis [15,16,17,18,19]. Our patient presented positivity of IgG against p83, p39, p30, p21, Osp17/DbpA, p14, and VlsE and of IgM against Osp17, VlsE, and p41 in serum, while in CSF he presented IgG against Osp17, p14, VlsE (lower intensity of bands in comparison with serum immunoblot), and totally negative for IgM (Figure 2). Moreover, PCR for Bb in CSF was negative. These features caused us to suppose no presence of acute LD but only a previous contact with Bb. IgG in CSF could be caused by high permeability of blood-CSF barrier due to the inflammation of the TBEV infection.

Diagnosis of TBEV infection could be based on clinical features in association with positive IgG and IgM antibodies in serum and in CSF (as our patient presented) [5]. Unfortunately, it was not possible to perform PCR for TBEV in CSF due to improper CSF storage [7].

## 4. Conclusions

Patients with suspicious cases of TBE may not be immediately tested for antibodies to TBEV in non-endemic regions, causing diagnostic delay of the disease. Clinician should consider this disease more as a differential diagnosis in patients with fever and neurological involvement, particularly in regions with high-tick density. This first autochthonous case supports the concept of circulation of TBEV and the presence of a possible TBEV hot spot in Serramazzoni, Modena. Further studies are required to establish guidelines for preventive measures such as vaccination in this region.

## Figures and Tables

**Figure 1 pathogens-11-00854-f001:**
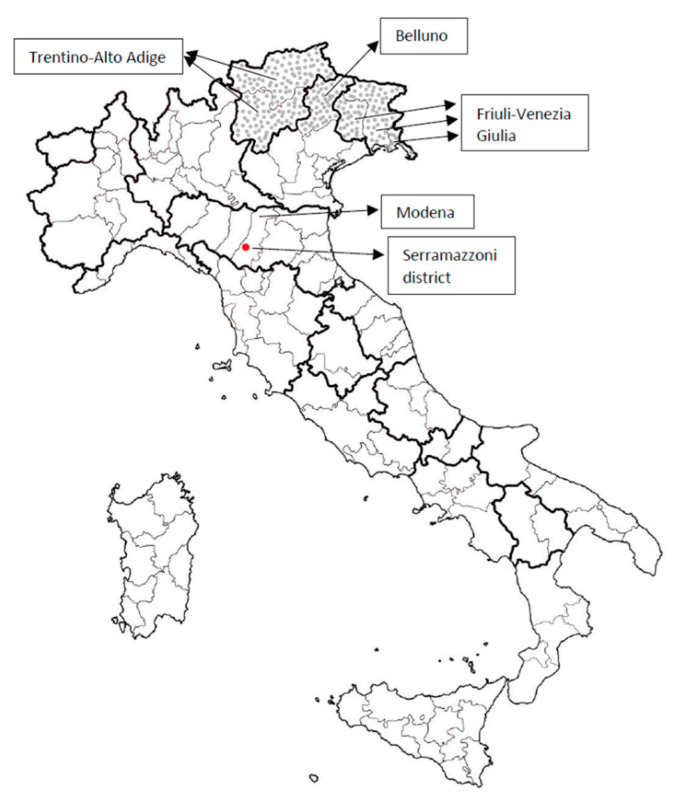
Endemic TBEV infection areas in the North-Eastern Italy are shown (Belluno, Trentino-Alto Adige and Friuli-Venezia Giulia). They are not bordering on Serramazzoni, the district of Modena, where the patient acquired TBEV.

**Figure 2 pathogens-11-00854-f002:**
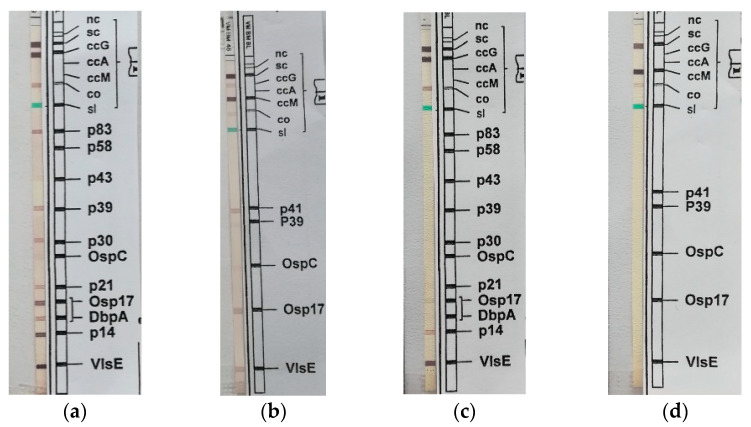
(**a**) IgG Immunoblot in serum; (**b**) IgM Immunoblot in serum; (**c**) IgG Immunoblot in CSF; (**d**) IgM Immunoblot in CSF. In serum IgG against p83, p39, p30, p21, Osp17/DbpA, p14, and VlsE and IgM against Osp17, VlsE and p41 are detected. In CSF IgG against Osp17, p14, VlsE were shown, but with lower intensity of bands in comparison with serum immunoblot. IgM in CSF were totally negative. Negativity of OspC (typical of previous LD), positivity of p83, p39, p21, and VlsE (typical of late immune response against Bb) and the lower intensity of immunoblot bands in CSF in comparison with immunoblot bands in serum induce to suppose previous contact with Bb.

**Table 1 pathogens-11-00854-t001:** Clinical features of the patient: after a first phase (beginning on 29 March) characterized by fever and asthenia, an asymptomatic interval was reported from 3 April to 11 April. Then, the second stage appeared with neurological involvement: primarily with fever, headache, ideomotor slowing, ataxia, constipation, high blood pressure levels, and dyspnea. On 15 April, tremors and facial palsy appeared. The patient was discharged on 21 April. After one month from the infection, tremors and asthenia remain present as sequelae.

Date	Event
29 March 2022	Fever, asthenia (First phase)
3 April 2022–11 April 2022	Asymptomatic interval
11 April 2022	Fever, headache, ideomotor slowing, ataxia, constipation, dyspnea, high blood pressure values (Second phase)
14 April 2022	Tremors, facial palsy
15 April 2022	Stop fever and dyspnea
21 April 2022	Discharged
6 May 2022	Tremors and asthenia persistence

## Data Availability

Not applicable.

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
