# Peer review of "First Human Case of Tick-Borne Encephalitis in Non-Endemic Region in Italy: A Case Report"

_pathogens, 2022, doi:10.3390/pathogens11080854_

Round 1
Reviewer 1 Report
importent for the TBE community as a new TBE focus is described
Author Response
Dear Reviewer 1,
Thank you for your revision
Reviewer 2 Report
Barp et al. describe a new autochronous case of serological diagnosed TBE in a region south of the known endemic TBEV regions in Nothern Italy.
The infection was diagnosed by serology, as the first part of the infection was not observed.
Some questions and suggestions to supplement the paper.
- How is this Serramazzoni area looks like? Vegetation cover, rainfall, humidity, altitude above see level?
- What is the distance of this Serramazzoni from the known nordic Italien TBEV endemic areas in km?
- Was the patient vaccinated before the illness?
- Did the patient consume goat-milk?
- Did the patient walk in bushy-forrested areas?
- Has the patient pets at home (cats, dogs)?
Why did not the authors attempt PCR from any of their biologicals samples from the patients? Even if the case was in its late phase?
Literature often supposes possible northward spread of the TBE virus because of the global warming problem. Here it spreads to South to more dry, more sunshined areas.
The paper could be supplemented with the answers of these questions.
Author Response
Dear Reviewer 2,
Thank you for your revision.
- How is this Serramazzoni area looks like? Vegetation cover, rainfall, humidity, altitude above see level?
Response: I describe topography and climate in the Discussion now: “…In this area the climate and availability of hosts are particularly suitable for the development of ticks and the maintenance of the life cycle of tick-borne pathogens Summers are warm and sunny, while winters are cold and snowy. During the year, the temperature generally ranges from -1 °C to 26 °C, rarely below -6 °C or above 30 °C. Average humidity is 75%, while rainfall ranges from about 78 millimeters in October to about 25 millimeters in January. In relation to topography, there are significant altitude variations (from 162 meters to 906 meters above sea level), with an average altitude of 669 metres; Serramazzoni is covered by tree areas (48%), cultivated land (46%) and different torrents. While bacteria belonging to genus Anaplasma, Rickettsia, Babesia and Borrelia were found in ticks (especially in Ixodes ricinus), TBEV-positive ticks had not been found in this district…”
- What is the distance of this Serramazzoni from the known nordic Italien TBEV endemic areas in km?
Response: I describe the distance in the Discussion now: “…Ticks which transmitted TBEV may have been transported by birds or other animals from endemic TBEV endemic regions (Figure 1), since the distance between Serramazzoni and the closest endemic area (Belluno), is about 300 kilometers…”
- Was the patient vaccinated before the illness?
Response: I add this information in The Case Description: “…He had no comorbidities, no chronic therapies and he did not remember any tick-bites; he was not vaccinated for TBE or Yellow Fever…”
- Did the patient consume goat-milk?
Response: I add this information in The Case Description: “…He had not consumed goat-milk and he had not travelled outside Serramazzoni, his home town, in the two months before becoming ill…”
- Did the patient walk in bushy-forrested areas?
Response: I add this information in The Case Description: “…However he went trekking in bushy and forested areas in this district…”
- Has the patient pets at home (cats, dogs)?
Response: I describe it in The Case Description: “…he was not vaccinated for TBE or Yellow Fever and he had not pets at home…”
- Why did not the authors attempt PCR from any of their biologicals samples from the patients? Even if the case was in its late phase?
Response: Firstly no one thought TBEV could be the responsible of the clinic, since patient had not been out of Serramazzoni, a district where no TBE cases had been registered before. This is the reason why we started testing immediately CSF for other viruses, with negative results. Only with negative results, we considered TBE and we sent CSF to Belluno’ Hospital where antibodies detection where performed with Enzyme-Linked Immunosorbent Assays (ELISA) “Vienna” test (with Analyzer I EUROIMMUN). For further investigation, also Reagena's ReaScan TBE IgM immunochromatographic test on CSF and on serum with automatic reading was used. However PCR for TBEV was not performed, due to improper CSF storage: it was not stored at -20°C. Our diagnosis was based on clinical features in association with positive antibodies for TBEV in serum and CSF as Eurosourveillance described (Beauté Julien, Spiteri Gianfranco, Warns-Petit Eva, Zeller Hervé. Tick-borne encephalitis in Europe, 2012 to 2016. Euro Surveill. 2018;23(45):pii=1800201)
Reviewer 3 Report
Manuscript written by Barp et al. important paper on the emergence of new foci of tick-borne encephalitis virus in Europe. The authors described the case report of tick-borne pathogens in Emilia-Romagna, Italy.
In Introduction is described little information about tick-borne encephalitis virus. Who is the main vector of the virus in Europe and in Italy, for example, the endemic areas of Italy? Where in Europe is tick-borne encephalitis virus spread in general and in Italy in particular? What tick-borne encephalitis virus subtypes are circulating in Europe and what can be expected from nearby countries? What subtype is circulating in Italy? What is the epidemiological situation of tick-borne encephalitis in the endemic areas of Italy?
Line 25: Flaviviridae must be italic.
Line 26-27: The territory may be endemic, the disease may be endemic to the area. The virus cannot be endemic.
The Case Description
In the Case Description the symptoms and tests performed are described in detail. But I have some comments and suggestions.
1. How long did the fever last in the first phase? Two, three, four or more days.
2. The authors tested CSF for various infections. Why wasn't CSF tested for West Nile and Dengue viruses? Italy is endemic for these diseases.
3. On 21st April, serology for TBEV was performed. Is this the first serum taken on admission, or are these subsequent. On what day from admission was the serum taken for analysis? Why was West Nile and Dengue not excluded if the test system also detects antibodies to these viruses?
4. There is no data on the patient's vaccine status. Was he vaccinated against tick-borne encephalitis or yellow fever? Has it been in flavivirus-endemic areas not in the last 2 months, but at least in a year?
5. Data from only one serum study are reported. Has the next sera been tested? Is there an increase in antibody titers against TBEV?
Line 39: tick bites without hyphen
Line 56: What liver enzymes are elevated? It may be associated with other diseases.
Discussion
Line 82: “This is the first autochthonous case of TBE…” no clear evidence for this.
Line 86: genus Anaplasma, Rickettsia, Babesia and Borrelia must be italic.
Line 88: endemic is a repeat.
Line 89: “Defining a TBE risk area in the emerging phase of the disease is difficult.” What did the authors mean by Defining a TBE risk area?
I can assume that this is indeed a case of tick-borne encephalitis. But the authors did not provide clear evidence that the patient had tick-borne encephalitis. The described clinic is similar to the clinic for tick-borne encephalitis. But one clinic does not make a diagnosis. Serum and CSF antibodies were detected by a test system that also detects antibodies to West Nile, Dengue, and Yellow Fever. But the study of serum and CSF for these viruses was not carried out. Why did the authors not examine serum and CSF using PCR for tick-borne encephalitis virus? The authors tested serum and CSF for Cytomegalovirus, Epstein Barr Virus, Herpes Virus 1 and 2, Adenovirus, Enterovirus, Toscana virus, different bacterial infections, and did not test for tick-borne encephalitis virus, although tick-borne encephalitis appears in the diagnosis. If the authors claim that this is the first case of tick-borne encephalitis, then where is the virus sequence, or at least the PCR result?
This article needs additional experiments.
Author Response
Dear Reviewer 3,
Thank you for your revision!
- In Introduction is described little information about tick-borne encephalitis virus. Who is the main vector of the virus in Europe and in Italy, for example, the endemic areas of Italy? Where in Europe is tick-borne encephalitis virus spread in general and in Italy in particular? What tick-borne encephalitis virus subtypes are circulating in Europe and what can be expected from nearby countries? What subtype is circulating in Italy?
Response: I added the information required: “…Three main virus subtypes are described: European or Western Tick-Borne Encephalitis Virus (TBEV-Eu), Siberian Tick-Borne Encephalitis Virus and Far Eastern Tick-Borne Encephalitis Virus. In Europe, including Czech Republic, Estonia, Latvia, Lituania, Poland, Switzerland, the Southern part of Scandinavian Peninsula, the North-Eastern part of Croatia, the Southern part of Germany, the Western and Northern part of Hungary and some districts of Austria the TBEV-Eu infection is endemic. In these areas, Ixodes ricinus act both as vector and reservoir…”
“…In Italy TBEV-Eu is endemic in three North-Eastern areas: Belluno, Trentino-Alto Adige and Friuli-Venezia Giulia…”
- Line 25: Flaviviridaemust be italic.
Response: “Flaviviridae” is change in “Flaviviridae”
- Line 26-27: The territory may be endemic, the disease may be endemic to the area. The virus cannot be endemic.
Response: I agree with you and I correct it: “…TBEV-Eu infection is endemic…”
The Case Description
In the Case Description the symptoms and tests performed are described in detail. But I have some comments and suggestions.
- How long did the fever last in the first phase? Two, three, four or more days.
Response: I addes this information in the text: “…The first one appeared on 29th March and was characterized by fever and asthenia for four days…”
- The authors tested CSF for various infections. Why wasn't CSF tested for West Nile and Dengue viruses? Italy is endemic for these diseases.
Response: Dengue is not endemic in Italy, there are only imported cases and extreme rare outbreaks in people who live near someone who got infected abroad. Regarding West Nile, we routinely test CSF for West Nile virus only when the virus is detected in sentinels (birds), usually from June to October. So, in our clinical case CSF was not tested for West Nile virus because in our area the virus had not been found in sentinels, yet. The patient did not travelled to other districts in the three months before. (Piano Nazionale di prevenzione, sorveglianza e risposta alle Arbovirosi (PNA) 2020-2025; https://www.salute.gov.it/imgs/C_17_pubblicazioni_2947_allegato.pdf)
- On 21st April, serology for TBEV was performed. Is this the first serum taken on admission, or are these subsequent. On what day from admission was the serum taken for analysis?
Response: I added this information: “…On 21st April (9 days from the admission), serology for TBEV both in serum and in CSF was performed…”. I confirm that it was the first serum taken on admission
- Why was West Nile and Dengue not excluded if the test system also detects antibodies to these viruses?
Response: see response above
- There is no data on the patient's vaccine status. Was he vaccinated against tick-borne encephalitis or yellow fever? Has it been in flavivirus-endemic areas not in the last 2 months, but at least in a year?
Response: I describe TBE vaccine status in The Case Description now: “…he was not vaccinated for TBE or Yellow Fever…”.
He has not been in flavivirus-endemic area in the last year. We decide to consider 2 months because incubation period of TBE is usually about 2-28 days. However in the last year the patient has not been in flavivirus-endemic areas.
- Data from only one serum study are reported. Has the next sera been tested? Is there an increase in antibody titers against TBEV?
Response: Other next sera have not been tested
- Line 39: tick bites without hyphen
Response: changed as suggested
- Line 56: What liver enzymes are elevated? It may be associated with other diseases.
Response: I describe it now: “…and liver elevation enzymes (especially ALT)…”
Discussion
- Line 82: “This is the first autochthonous case of TBE…” no clear evidence for this.
Response: no cases of TBE had been found before in this district. The patient did not travelled outside Serramazzoni district in the months before the diagnosis, but he went trekking in forests of the district. Diagnosis was based on clinical features in association with positive antibodies for TBEV in serum and CSF as described in Eurosourveillance (Beauté Julien, Spiteri Gianfranco, Warns-Petit Eva, Zeller Hervé. Tick-borne encephalitis in Europe, 2012 to 2016. Euro Surveill. 2018;23(45):pii=1800201)
- Line 86: genus Anaplasma, Rickettsia, Babesiaand Borrelia must be italic.
Response: “Anaplasma, Rickettsia, Babesia and Borrelia” change in “Anaplasma, Rickettsia, Babesia and Borrelia” as suggested.
- Line 88: endemic is a repeat.
Response: I changed “from endemic TBEV endemic regions” to “from TBEV endemic regions”
- Line 89: “Defining a TBE risk area in the emerging phase of the disease is difficult.” What did the authors mean by Defining a TBE risk area?
Response: with “TBE risk area” I mean a region where TBE could be an emerging disease and vaccine could be necessary. Actually only our case is reported, but the presence of Ixodes ricinus do not exclude a significant emergence of TBE in this district, defining it a risk area for TBE.
- I can assume that this is indeed a case of tick-borne encephalitis. But the authors did not provide clear evidence that the patient had tick-borne encephalitis. The described clinic is similar to the clinic for tick-borne encephalitis. But one clinic does not make a diagnosis. Serum and CSF antibodies were detected by a test system that also detects antibodies to West Nile, Dengue, and Yellow Fever. But the study of serum and CSF for these viruses was not carried out. Why did the authors not examine serum and CSF using PCR for tick-borne encephalitis virus? The authors tested serum and CSF for Cytomegalovirus, Epstein Barr Virus, Herpes Virus 1 and 2, Adenovirus, Enterovirus, Toscana virus, different bacterial infections, and did not test for tick-borne encephalitis virus, although tick-borne encephalitis appears in the diagnosis. If the authors claim that this is the first case of tick-borne encephalitis, then where is the virus sequence, or at least the PCR result?
Response: Firstly no one thought TBEV could be the responsible of the clinic, since patient had not been outside of Serramazzoni, a district where no TBE cases had been registered before. This is the reason why we started testing immediately CSF for other viruses, with negative results. Only with negative results, we considered TBE and we sent CSF to Belluno’ Hospital where antibodies detection where performed with Enzyme-Linked Immunosorbent Assays (ELISA) “Vienna” test (with Analyzer I EUROIMMUN). For further investigation, also Reagena's ReaScan TBE IgM immunochromatographic test on CSF and on serum with automatic reading was used. However PCR for TBEV was not performed, due to improper CSF storage: it was not stored at -20°C. Our diagnosis was based on clinical features in association with positive antibodies for TBEV in serum and CSF as Eurosourveillance described (Beauté Julien, Spiteri Gianfranco, Warns-Petit Eva, Zeller Hervé. Tick-borne encephalitis in Europe, 2012 to 2016. Euro Surveill. 2018;23(45):pii=1800201).
Round 2
Reviewer 3 Report
The authors answered all my questions and made some additions to the text. But some of the information was left outside the manuscript.
1. When you submit an article to an international journal, you understand that it will be read by scientists from different countries and that the criteria for confirming or refuting a disease in different countries are different. The criteria by which you confirm the diagnosis of tick-borne encephalitis should be in the introduction, and not in the last sentence of the discussion as a reference.
2. You response: “Regarding West Nile, we routinely test CSF for West Nile virus only when the virus is detected in sentinels (birds), usually from June to October. So, in our clinical case CSF was not tested for West Nile virus because in our area the virus had not been found in sentinels, yet.” According to your reference (Piano Nazionale di prevenzione, sorveglianza e risposta alle Arbovirosi (PNA) 2020-2025) you should pay attention to WNV during the period of increased vector activity (May-November). But the average temperature in the region since February has already allowed mosquitoes to hatch. Therefore, we cannot exclude WNV. In any case, omitting the above, the text of the article does not mention anywhere why you excluded WNV from diagnostics.
3. You response: “He has not been in flavivirus-endemic area in the last year.” According to your reference (Piano Nazionale di prevenzione, sorveglianza e risposta alle Arbovirosi (PNA) 2020-2025) Emilia Romagna is endemic for WNV, therefore, the patient lives in a flavivirus endemic region and may have antibodies against flaviviruses.
4. You response: “However PCR for TBEV was not performed, due to improper CSF storage: it was not stored at -20°C.” You will not be able to isolate the virus in cell cultures, because infectious virions died. However, you can isolate the RNA because it remains unchanged, even if stored incorrectly. And you also have blood serum, from which you can also extract RNA.
Author Response
Dear reviewer, thank you for your revision:
- When you submit an article to an international journal, you understand that it will be read by scientists from different countries and that the criteria for confirming or refuting a disease in different countries are different. The criteria by which you confirm the diagnosis of tick-borne encephalitis should be in the introduction, and not in the last sentence of the discussion as a reference.
Response: I added the information required in the Introduction: “European Union case definition of TBE is based on symptoms of inflammation of the Central Nervous System and one of the following laboratory confirmation: TBEV specific IgG and IgM antibodies in blood, TBEV specific IgM antibodies in CSF, seroconversion or fourfold increase of TBE specific antibodies in paired serum samples, detection of TBEV viral nucleic acid or isolation of TBEV from clinical specimen.”
- You response: “Regarding West Nile, we routinely test CSF for West Nile virus only when the virus is detected in sentinels (birds), usually from June to October. So, in our clinical case CSF was not tested for West Nile virus because in our area the virus had not been found in sentinels, yet.” According to your reference (Piano Nazionale di prevenzione, sorveglianza e risposta alle Arbovirosi (PNA) 2020-2025) you should pay attention to WNV during the period of increased vector activity (May-November). But the average temperature in the region since February has already allowed mosquitoes to hatch. Therefore, we cannot exclude WNV. In any case, omitting the above, the text of the article does not mention anywhere why you excluded WNV from diagnostics.
Response: I added this information as suggested: “West Nile virus was not tested since in Emilia Romagna the virus is usually tested only during the period of increased vector activity (May-November) and was not already been detected in sentinels (birds) in that period of the year [6].”
- You response: “He has not been in flavivirus-endemic area in the last year.”According to your reference (Piano Nazionale di prevenzione, sorveglianza e risposta alle Arbovirosi (PNA) 2020-2025) Emilia Romagna is endemic for WNV, therefore, the patient lives in a flavivirus endemic region and may have antibodies against flaviviruses.
Response: You are right, sorry for the misunderstanding. I meant that the patient has not been in an area endemic for TBE or yellow fever in the last year
- You response: “However PCR for TBEV was not performed, due to improper CSF storage: it was not stored at -20°C.” You will not be able to isolate the virus in cell cultures, because infectious virions died. However, you can isolate the RNA because it remains unchanged, even if stored incorrectly. And you also have blood serum, from which you can also extract RNA.
Response: We have only blood serum of the patient during the neurological involvement of the disease (second phase), so we cannot extract TBEV-RNA in blood, since it could be found during the first phase (viremic phase) of the disease. Regarding CSF, we follow recommendations of Deisenhammer et al. for investigation of infectious CSF (Deisenhammer F, Bartos A, Egg R, Gilhus NE, Giovannoni G, Rauer S, Sellebjerg F; EFNS Task Force. Guidelines on routine cerebrospinal fluid analysis. Report from an EFNS task force. Eur J Neurol. 2006 Sep;13(9):913-22. doi: 10.1111/j.1468-1331.2006.01493.x. PMID: 16930354): cerebrospinal fluid should be immediately (i.e. <1 h) analyzed after collection, but if storage is required for later investigation, this can be done at 4–8°C (short term, 24-48h) or at -20°C (long term). Since at first no one thought about TBEV, CSF was not tested for TBEV in short term; when we sent the CSF to the TBEV referral laboratory, the lab did not perform PCR due to the improper CSF storage.